# Practical Aspects of the Use of Telematic Systems in the Diagnosis of Acute Coronary Syndrome in Poland

**DOI:** 10.3390/medicina58040554

**Published:** 2022-04-17

**Authors:** Lukasz Gawinski, Monika Burzynska, Karolina Kamecka, Remigiusz Kozlowski

**Affiliations:** 1Department of Management and Logistics in Health Care, Medical University of Lodz, 90-237 Lodz, Poland; karolina.kamecka1@stud.umed.lodz.pl; 2Department of Epidemiology and Biostatistics, Medical University of Lodz, 90-237 Lodz, Poland; monika.burzynska@umed.lodz.pl; 3Department of Emergency Medicine and Disaster Medicine, Medical University of Lodz, 90-237 Lodz, Poland; remigiusz.kozlowski@umed.lodz.pl

**Keywords:** myocardial infarction, transtelephonic electrocardiography, decision making process

## Abstract

*Background and Objectives*: The guidelines of the European Society of Cardiology (ESC) recommend the use of telematic methods in the diagnosis of myocardial infarction, allowing for transtelephonic electrocardiography (TTECG) from the emergency scene to centers performing percutaneous coronary interventions (PCI center). It has been proven that such a procedure has a beneficial effect on the survival of patients with ST elevation myocardial infarction (STEMI). Fewer data can be found on the correct use of these methods in everyday clinical practice. The aim of this study was to indicate potential indications and contraindications for the use of the TTECG system, and provide recommendations for proper collaboration between emergency medical systems (EMS) teams and PCI centers. *Materials and Methods*: The article is a systematic review of cardiological emergencies, with an assessment of indications for the use of the TTECG system. The authors introduced their own grading of the validity of indications for transmission of the TTECG, similar to those used in the official ESC guidelines.: *Results:*: The authors described individual cardiological emergencies occurring in the practice of EMS, considering specific indications or contraindications for the transmission of the TTECG. The article also discusses individual practical recommendations for proper cooperation between EMS teams and PCI centers in detail. All of the recommendations are compiled in a handy table to facilitate its use in everyday clinical practice. *C**onclusions*: The summary presents a comparison of the realities of the functioning of the telematic support system in Poland in the field of STEMI diagnostics, with the model’s recommendations. The necessity of further educating the members of individual teams included in the network dealing with STEMI treatment was indicated, as well as the necessity of introducing legal regulations sanctioning the functioning of telematic systems in modern medicine.

## 1. Introduction

Modern medicine is frequently based on the latest technologies, ranging from pharmacological [1], through to medical robotics [2], and to modern techniques in the field of telemedicine [3,4,5]. The latter have recently played an increasingly significant role and have created new, previously unattainable possibilities for supporting the diagnosis and treatment of patients. One example of the use of information technology in medicine is the use of telematic techniques, which are often used in modern emergency medicine [6]. Telematics is defined as the process of manipulating information and data. This applies to the acquisition, processing, distribution, transmission, and use of these data in a variety of decision-making processes [7]. Due to the nature of work in an emergency medical system (EMS) (operating in the field outside of the hospital structure, with no possibility of directly consulting a specialist in a given field, with limited time and diagnostic methods, and highly urgent events), telematic systems allow workers to send various clinical data of the patient and obtain help for further proceedings. This has provided essential and valuable support for work in EMS. An excellent example illustrating these assumptions is the possibility of transmitting transtelephonic electrocardiography (TTECG) for a patient suspected of having a myocardial infarction (MI) from the scene of the event to centers performing percutaneous coronary interventions (PCI center) in order to support the decision-making process regarding the qualification of the patient for urgent coronary angioplasty [8,9]. It should be mentioned here that the guidelines of the European Society of Cardiology (ESC) clearly recommend the use of telematic systems in everyday clinical practice to support the process of diagnosis and treatment of ST elevation myocardial infarct (STEMI) [10]. In the contemporary literature, there are reports on organizing the treatment of STEMI patients in individual countries [11,12], as well as descriptions of the telematic systems used to optimize and support the treatment process of these patients [13]. The use of telematic methods allows workers, as recommended in the ESC guidelines, to transport patients with STEMI directly to a PCI center (intentionally bypassing hospitals without the ability to perform PCI, i.e., a non-PCI center). When a patient with STEMI is initially transported to a non-PCI center, it is estimated that the median time spent in this hospital before transfer to a PCI center is approximately 68 min [14]. This delay is related to the process of diagnostics and organizing further treatment (contact with the PCI center and organizing medical transport). The same study proved that waiting times of a patient in a non-PCI center for transport exceeding 30 min were closely related to an increase in in-hospital mortality in these patients [14]. Another benefit is the ability to transport the patient directly to the catheterization laboratory (CL), bypassing the emergency department (ED), which shortens the time from first medical contact (FMC) to wire crossing by 20 min [15]. These capacities clearly indicate that the use of a TTECG system directly translates into a reduction in the time from the FMC to reperfusion [16,17], and has a fundamental beneficial effect on the prognosis of a patient with STEMI [18,19].

Currently, the classification of MI is divided into two main divisions. The first division is based on the pathophysiological mechanisms responsible for the development of MI that have been presented in the fourth general definition of MI [20]. This classification is based on the results of additional tests (coronary angiography results, laboratory tests results), and is not applicable in the pre-hospital phase. The second classic division of MI is based on the ECG result. In outpatient settings, a diagnosis of MI is based primarily on the clinical signs (location, nature, and type of chest pain), and on the ECG results. Ambulances of EMS are equipped with portable ECG devices as standard (most often as part of a mobile defibrillator), which allows the first responders to perform an ECG at the scene (at the patient’s home, in a public place). It should be remembered that carrying out an ECG on a patient with chest pain (with suspected MI) is recommended by the ESC within 10 min of FMC with the patient [10]. A discussion of the precise electrocardiographic criteria for a diagnosis of STEMI of the individual walls of the heart, the criteria for a diagnosis of STEMI in the left/right Hiss bundle branch block, or in the presence of paced rhythm significantly exceeds the scope of this study, and has been thoroughly described in the literature [10,21,22,23]. Contemporary STEMI treatment is based on reperfusion treatment methods that allow for restoration of the patency of a closed coronary artery. There are two methods of reperfusion treatment: pharmacological reperfusion and mechanical reperfusion. Pharmacological reperfusion, also called fibrinolytic therapy, involves the intravenous administration of drugs to dissolve the clot that blocks the flow in the coronary artery. Mechanical reperfusion is the mechanical opening of the coronary artery with a balloon, stent, or other device, and is referred to as primary coronary angioplasty (PCA). The superiority of PCA over fibrinolytic therapy in patients with STEMI should be clearly emphasized [24]. An exception is when a significant delay in transporting the patient to the CL of more than 2 h is expected, in which case the administration of an intravenous bolus of a fibrinolytic drug at the scene of the event should be considered. The authors’ own experience shows that another common indication for the administration of fibrinolysis is the patient’s refusal to accept invasive treatment. Fibrinolysis is then an alternative to mechanical reperfusion, but the patient should be warned about bleeding complications and worse treatment outcomes [25]. In line with the ESC guidelines [10], it is recommended to create STEMI treatment systems within individual countries. The system should include a nationwide network of CL with the possibility of performing PCA (24 h a day, 7 days a week), operating on the basis of highly specialized cardiology departments. These departments are located in hospitals with adequate facilities for the treatment of MI (intensive care units, local EDs). The STEMI treatment system also includes smaller local hospitals that do not have a PCI center in their structure, and a well-functioning EMS that combines all these elements. The proper operation of all three of these components ensures the proper functioning of the entire system, and guarantees the best care for patients with MI. In 2020, there were 154 CLs in Poland, where 575 certified independent invasive cardiology operators were working [26]. Details on the structure, functioning, and characteristics of EMS in Poland have been described in detail in the literature [13]. It should be emphasized once again that the main task of telematic support systems in the form of TTECG systems is the diagnosis and qualification for invasive treatment of patients with STEMI or high-risk NSTEMI.

## 2. Aim of the Work

This work is an attempt to create practical recommendations for the use of the TTECG system in everyday clinical practice and is a systematic review of cardiological emergencies with an assessment of the indications for the use of the TTECG system. The article also provides recommendations for collaboration between EMS teams and PCI centers. It should be remembered that the following recommendations were made only on the basis of the professional experience of the authors, and constitute a subjective point of view of the authors of the work, and are not supported by clinical studies or clinical registries. However, clinical experience (working in EMSs, in PCI centers, and in the CL) has allowed the authors a unique and universal approach to the issues of cooperation between the EMS and PCI centers. Due to the complexity of the issues and the enormous number of factors that should be considered when using telematic systems, it is not always possible to indicate a clear mode of action. The authors have tried to introduce their own system of grading the importance of individual recommendations, which is analogous to and based on the classes of recommendations used in the ESC guidelines [10]. Since these are only the authors’ opinions and have not been supported by clinical trials, the level of evidence of the following recommendations is at the level of C. This publication takes particular account of the realities of working in the Polish public health care system.

## 3. Cooperation between the EMS and the PCI Center

The next part of the article discusses particularly important aspects of mutual cooperation during the use of a TTECG system between the EMS and the PCI center. The theoretical assumptions of the system seem to be simple and clear. Unfortunately, everyday life and clinical practice have shown that during operations, many unclear, disputable, and conflicting situations arise, which directly translate into an increase in the time from the FMC to wire crossing, which, in turn, directly translates into poorer survival outcomes in patients with STEMI. 

One of the principal issues in the recommendations for the ESC in the guidelines is the introduction of clearly defined geographical areas of responsibility for a given PCI center. Each EMS team, working on its own premises, should cooperate with the PCI center assigned to it on a permanent basis (suggested class of recommendations: I). Of course, this assignment depends on the distance to a PCI center. Members of the EMS should not be free to choose the PCI center to which they will transmit the TTECG. This is conducive to improving the conditions of cooperation, good knowledge of contact numbers, and reduces the risk of adverse events related to the organization and logistics of the treatment process of patients with STEMI. According to the ESC guidelines for the treatment of STEMI [10], all rules and principles of cooperation should be created in the form of simple written protocols that should be read and followed by all members of the individual teams included in the STEMI treatment system (suggested class of recommendations: I). This facilitates subsequent cooperation and conflict resolution. A lack of clearly formulated and written protocols favors individual interpretations of events in a way that is favorable to one of the sides of a potential problem, which may not always be correct, and is often not beneficial to the patient. It should be remembered that all of the introduced rules of cooperation are designed to serve only the good of the patient and provide her/him with the fastest and best medical assistance using the resources and possibilities available at the moment.

One example of the lack of cooperation is the transportation of a patient to a distant PCI center, despite the fact that the cardiologist on duty did not find any indications for transport to a PCI center (suggested class of recommendations: III). As is well known, there are many causes of chest pain. The literature includes articles analyzing patients admitted to the ED with chest pain, revealing that approximately 75% to 85% of patients with chest pain are not ultimately diagnosed with MI [27]. Direct transport of a patient with chest pain to a PCI center is always necessary only in the case of patients with STEMI. In the absence of features of a STEMI infarction on the ECG in a patient with chest pain, non-ST elevation myocardial infarct (NSTEMI) may be suspected. In this group of patients, very high-risk patients who qualify for an immediate invasive strategy are characterized by: hemodynamic instability, cardiogenic shock, recurrent/refractory chest pain despite pharmacological treatment, life-threatening arrhythmias, mechanical complications of MI, acute heart failure clearly associated with NSTEMI, ST segment depression >1 mm in six leads, and ST segment elevation in a VR and/or V1 [28]. These groups of patients should also be immediately transported to a PCI center (despite the absence of STEMI features in the ECG). The remaining patients may be transported to the ED of the nearest hospital, where they will be given the necessary medical assistance and appropriate diagnostic procedures will be conducted. After confirming NSTEMI, in accordance with the risk assessment carried out, these patients can be transported to a PCI center within appropriate timeframes, in accordance with the ESC guidelines. Of course, this rule does not apply to patients for whom the hospital with the PCI center is the nearest hospital. A priori transport of each patient with chest pain to a PCI center may very often be harmful to patients (it extends the patient’s transport time to the appropriate medical center in the case of patients who have not finally been diagnosed with MI), and may result in a higher risk of death (suggested class of recommendations: III). Secondly, this situation favors a significant overcrowding of the ED and PCI centers, and a significant workload for their staff. It should be remembered that PCI centers cooperate with several or a dozen smaller hospitals, and look after patients living in a large area. Therefore, one should always account for the correct use of bed resources and highly specialized personnel working in these centers, so that, at the critical moment, there is no shortage of a bed for a patient with confirmed STEMI.

Another exception is the refusal to admit a patient with a STEMI for technical reasons (e.g., angiograph failure, other STEMI patients awaiting a procedure). In such a case, the patient should absolutely not be transported to this PCI center (suggested class of recommendations: III). Such a procedure may significantly extend the time until implementation of interventional treatment, and result in a worsening of the patient’s prognosis or even her/his death. At this point, it should also be noted that if the CL assigned to a given ambulance team refuses to admit a patient with STEMI for the abovementioned reasons, she/he should indicate the nearest CL that is able to admit and provide for such a patient.

## 4. Indications and Contraindications for TTECG Transmission to a PCI Center

Other practical aspects worth discussing are the indications for transmission of an TTECG. At this point, it should be clearly emphasized that the only indication for transmission of the TTECG leads to the suspicion of a MI (in presumed a STEMI). Therefore, the only indisputable indications for TTECG transmission will be potential symptoms of a MI, which most often comes down to the patient reporting chest pain (suggested class of recommendation: I). Undoubtedly, the unnecessary and unjustified use of the TTECG system is a problem in everyday clinical practice. Abdominal pain, dizziness, disturbance of consciousness, malaise in general, and patients with chest trauma after a traffic accident are the most frequent causes of the TTECG transmission in Poland. It must be clearly emphasized that the possibility of TTECG transmission does not lead to the possibility of the EMS team consulting with the cardiologist on duty about every problematic patient. In the abovementioned cases, the time spent on transmitting the TTECG is time wasted, because in these cases, emergency procedures, which may impact on the patient’s survival, must be undertaken immediately. Excessive unjustified TTECG transmission may lead to overloading the cardiologist on duty with work and, at certain times, may significantly disrupt the proper functioning of PCI center. 

Gastric symptoms such as abdominal pain, nausea, and vomiting are quite common causes of TTECG transmission. There are reports in the literature on how MI mimics abdominal pain [29]. The authors’ own experience shows that the so-called abdominal mask of MI is a relatively rare phenomenon in everyday clinical practice, and is most often associated with massive ischemia of the lower heart wall. Therefore, an acute abdominal condition or another acute surgical condition should always be suspected first in a patient reporting abdominal pain. As part of differential diagnosis, one should always focus on the most likely diagnosis first, and move gradually to the rarer and less probable ones. Peritoneal signs, abdominal tenderness, and a lack of intestinal peristalsis are not symptoms related to the abdominal mask, and unequivocally suggest an acute state of the abdominal cavity. In case of doubt, transmission of the TTECG recording should be considered (suggested class of recommendation: IIa).

Irrespective of its origin, bradyarrhythmia is generally not a classic indication for TTECG transmission. One of the exceptions to this rule is a diagnosis of acute MI complicated by a complete atrioventricular block or sinus bradycardia. However, in this case, the first symptom most frequently reported by the patient is chest pain, which initially suggests the diagnosis of MI, and indicates transmission of the TTECG. It is worth noting that a complete atrioventricular heart block is quite a rare complication of STEMI: it occurs in approximately 1.7% of patients with STEMI of the anterior wall and in 10.7% of patients with non-anterior STEMI [30]. It should be clearly emphasized that a diagnosis of bradycardia by the EMS team at the scene without the accompanying typical pain in chest is not a classic indication for TTECG transmission (suggested class of recommendation: IIb). Of course, an ECG should be made to identify the cause of the bradycardia. The situation when members of the EMS team, for distinct reasons, are not able to correctly recognize the cause of bradycardia in the ECG (e.g., atrioventricular blocks, sinus bradycardia, sinus arrest, etc.) is also not an indication for transmission of the TTECG to a PCI center. Knowledge of the causes of bradycardia at the pre-hospital stage does not influence further medical management. In symptomatic bradycardia (regardless of the causes), standard pharmacological treatment should be promptly administered. In some cases, percutaneous pacing of heart should be considered. The EMS members should avoid wasting time for TTECG transmission (and waiting for feedback), and should immediately focus on implementing appropriate treatment at the scene. Unnecessary delays in undertaking appropriate emergency procedures related to the unnecessary transmission of the TTECG may have a negative impact on the patient’s chances of survival. Patients with life-threatening bradycardia, after initial treatment at the scene, should be taken to the ED of the nearest hospital, where temporary transvenous heart pacing can be implemented.

In case of patients with tachyarrhythmias, the situation is more complex. The group of stable patients with non-life-threatening supraventricular arrhythmias and the group of patients with ventricular tachycardia should be clearly distinguished here. It should be remembered that for both patients from the first and the second groups, the priority for the EMS team members should be the treatment of arrhythmias in accordance with the accepted standards. Transmission of the TTECG is a secondary activity; it should not delay the treatment of arrhythmias, and should take place after the patient’s clinical condition has been fully stabilized. It should be emphasized once again that the possibility of contacting the cardiologist on duty does not constitute the possibility to consult her/him regarding the methods of treating arrhythmias or the type and doses of the administered medications, and should not be used for telephone consultations on indications for electric cardioversion procedures. Such decisions can only be made by the members of the EMS at the scene after examining the patient personally. The most common narrow-QRS arrhythmia that occurs in the course of MI is atrial fibrillation, with a reported incidence of between 10% and 20%, depending on the type of MI [31]. In the case of hemodynamically stable supraventricular arrhythmias (with narrow QRS complexes) that proceed without obvious typical pain in the chest, there are no absolute indications for transmitting the ECG to a PCI center (suggested class of recommendation: IIb). In these patients, it should be assumed that the risk of a given arrhythmia being a direct complication of acute MI is low. These patients require transport to the nearest ED for further treatment. In patients with hemodynamically unstable supraventricular tachycardia and typical stenocardial symptoms, after clinical stabilization at the scene (which, in practice, means attempting to restore the sinus rhythm), transmission of the TTECG can be performed (suggested class of recommendation: IIa).

In the case of ventricular arrhythmias (VA) (in simple terms, arrhythmias with wide QRS complexes), the situation is simpler, since every patient with VA requires further detailed cardiological diagnostics, and should be referred to a specialized center. It is estimated that 6–8% of patients with STEMI have hemodynamically significant VA [32]. In this case, after the initial stabilization of the patient’s condition (which most often amounts to an attempt to restore the sinus rhythm), the ECG should be re-recorded and sent for analysis to a PCI center (suggested class of recommendations: IIa). It should be emphasized here that in the case of VA, analysis of the ST segment in the ECG is unreliable, and thus sending an ECG record in which tachycardia with wide QRS has been diagnosed is pointless, and reduces the patient’s chances of survival due to the loss of time devoted to this activity (suggested class of recommendation: III). It should also be remembered that for patients with VA who fail to restore sinus rhythm at the scene and are transported with an ongoing arrhythmia, the duration of transport should be minimized. In practice, this means transporting such a patient to the nearest ED (this time should not be extended by transporting the patient to a distant PCI center). In the ED after clinical stabilization of the patient (restoration of sinus rhythm), contact with the PCI center and transmission of the TTECG may be considered (suggested class of recommendations: IIa).

Dyspnea is another symptom that causes frequent TTECG transmission. The GRACE registry found that chest pain was absent in 8% of MI patients, while dyspnea was the main symptom in more than half of MI patients [33]. For the purposes of this study, the causes of dyspnea can be broadly divided into two groups: causes related to the respiratory system, and causes related to the cardiovascular system. The indication for transmission of the TTECG is a patient with severe dyspnea corresponding to the clinical features of pulmonary edema, in whom acute MI is the presumed cause of dyspnea. Chest pain, sudden onset, severe dyspnea, features of hemodynamic instability, hypotension, and features of pulmonary circulation congestion on physical examination are symptoms that support the cardiogenic background of an episode of dyspnea, and suggest transmission of the TTECG in order to exclude or confirm a possible STEMI (suggested class of recommendation: I). In everyday clinical practice, in the case of pulmonary edema, only the presence of a STEMI in the ECG is an indication for immediate invasive strategies. In many cases, it is necessary to start mechanical ventilation of the patient, because a patient with severe dyspnea is most often unable to assume the stable long-term lying position that is necessary for performing coronary angiography. In the case of a STEMI, the therapeutic priority is to restore the patency of the closed coronary artery. Therefore, the time to perform PCA should not be extended by devoting it to attempts to stabilize the patient and reduce dyspnea (to the extent that allows the patient to lie down). Such a procedure, although it seems to be correct at a first glance (the intention to avoid the need for endotracheal intubation and mechanical ventilation of the patient), reduces the patient’s chances of survival overall due to the significant extension of the time needed to restore the patency of the infarct artery. Although the ESC guidelines also recommend an immediate invasive strategy (up to 2 h) in NSTEMI patients with features of acute heart failure clearly associated with MI (with clinical features of pulmonary edema), in clinical practice, however, before making such important decisions regarding the patient (endotracheal intubation, implementation of mechanical ventilation), one should always try to confirm the diagnosis of MI (e.g., by measure cardiac troponins with high-sensitivity assays). The clinical presentation and the ECG itself are often not convincing enough for a diagnosis of NSTEMI (as it is known that NSTEMI may occur with minimal or non-specific changes in the ECG). Dyspnea in patients with a history of chronic lung disease, dyspnea in patients with fever with suspected pneumonia, dyspnea in patients not presenting typical pain in the chest, and dyspnea in hemodynamically stable patients are not indications for transmission of the TTECG (suggested class of recommendation: III). An additional difficulty in the case of a patient with serious dyspnea are technical problems with performing the correct ECG (the patient is agitated and often takes a sitting position, non-cooperative patients), which contributes to a large number of artifacts in the ECG and significantly reduces its diagnostic value. In these cases, the on-duty doctor receiving the ECG in the PCI center will not be able to correctly assess the ECG. In such a situation, time should not be wasted on repeating the ECG several times (especially in the case of a patient in a severe clinical condition). In the case of dyspnea, the priority is to implement medical procedures to save the patient’s life, not to focus on performing the ECG correctly.

The next group of patients are patients with a subcutaneously implanted cardioverter–defibrillator (ICD) after shock therapy. Each patient of this type should be examined in a specialized center in order to confirm the adequacy of the shock therapy, and then to determine the causes of arrhythmia. One of the possible and fairly frequent causes of VA triggering an ICD intervention in clinical practice is acute MI [34]. It should be remembered that a substantial proportion of patients with ICD have suffered from coronary heart disease or MI in the past. Therefore, the risk of MI in these group of patients is high. Upon arrival at the scene and the stabilization of the patient’s condition, the ESC team should transmit the TTECG to a PCI center (suggested class of recommendation: I). Particular attention should be paid to patients with recurrent arrhythmia, those with consecutive ICD shocks (with symptoms of an electrical storm) [35], those who report chest pain, and patients with hypotension (with suspected cardiogenic shock). In this case, it should also be remembered that emergency medical procedures always have priority, and transmission of the TTECG should not delay these. It should be emphasized that transmission of the TTECG does not automatically qualify the patient for urgent transfer to a PCI center. In the case of patients with no signs of hemodynamic instability, no recurrent arrhythmias, and no features of STEMI in ECG, there is no need for immediate transfer to a PCI center.

Syncope, defined as a transient, spontaneous loss of consciousness, is a common cause of ECG transmission to a PCI center. It should be noted that fainting is not a typical symptom of acute MI, and is not a typical indication for transmission of the TTECG. The causes of syncope are truly diverse, and are not related to cardiovascular disorders only. However, there are reports in the literature that an atypical clinical picture in the form of syncope was found in 10.6% of patients with acute cardiac ischemia [36]. Of course, tachyarrhythmias and bradyarrhythmias should always be considered as causes of syncope; however, these disorders are most often temporary. The EMS team arriving at the scene of the incident most often meets a patient who has already regained consciousness; therefore, the ECG does not show any disorders responsible for fainting. A patient who, after syncope, does not present with stenocardia and is hemodynamically stable has no obvious indications for transmission of the TTECG to a PCI center (suggested class of recommendations: IIb). Obviously, this recommendation does not mean that such a patient has no indications for an ECG test. In any other case, especially in patients with recurrent syncope, with typical stenocardial pain and with symptoms of hemodynamic instability, transmission of the TTECG to a PCI center should be considered in order to assess the indications for urgent PCA (suggested class of recommendations: IIa).

Patients in a coma constitute a separate group. A coma is defined as a state of prolonged unresponsive unconsciousness, and may be caused by various diseases. The coma may have a structural, metabolic, toxicological, or infectious etiology [37], and is associated with damage to the functions of the central nervous system. It should be clearly emphasized that coma is not a typical symptom of MI. Unconscious patients require standard emergency procedures with assessment of their vital signs. Coma in a patient with normal blood pressure and normal heart rate values is not a symptom of cardiovascular system disorders, and clearly indicates neurological causes. Transmission of the TTECG to a PCI center for such patients is not recommended (suggested class of recommendation: III).

Patients after successful resuscitation in the course of sudden cardiac arrest (SCA) constitute a very heterogeneous group, and are a great challenge for modern medicine. In the case of these patients, it is also important to remember to focus on emergency procedures first. Not every SCA survivor has to be routinely transported directly from the scene to a PCI center (suggested class of recommendations: III). According to the recent ESC guidelines on NSTEMI [28], only SCA survivors who have the features of a STEMI on the ECG are absolute candidates for an immediate invasive strategy. In the case of patients with NSTEMI, the superiority of a routine immediate invasive strategy over a deferred invasive strategy has not been proven [38]. For this reason, members of EMS should transmit the TTECG to a PCI center after successful reanimation (suggested class of recommendations: I). SCA survivors without the features of STEMI in ECG should be transferred to the nearest hospital with an intensive care unit (ICU). Long-term transport of a patient without indications of STEMI to a distant PCI center should be treated as a mistake that may reduce the patient’s chances of survival. It should be remembered that such patients, after the necessary tests have been performed in the nearest hospital, may be candidates for an early invasive strategy (i.e., transfer to a PCI center and performing coronary angiography within 24 h of cardiac arrest). Standard transfer of each SCA survivors to the PCI center also causes quick occupation of free beds in the intensive care unit. Patients who, after the diagnosis, have no indications for coronary angiography may remain in this hospital anyway (due to the potential risk of complications or death during transfer) and not be transported back to their home hospitals, where they should originally have been admitted after the SCA incident. Such a procedure very quickly leads to the occupation of all free beds in the ICU, and makes it impossible to admit more patients requiring invasive treatment.

All of the recommendations in the article regarding the use of telematic systems to support the diagnostic process of MI described above are summarized in Table 1.

## 5. Conclusions

The introduction of telematic methods has undoubtedly significantly improved the process of diagnosis and treatment of MI, which indirectly translates into better treatment results. As mentioned, the use of telematics is officially recommended in the ESC guidelines. Unfortunately, the same guidelines do not indicate further practical rules and principles allowing for the optimal use of this technology in everyday medical practice. The above considerations clearly indicate that the incorrect use of TTECG system may cause more losses than gains, and translate not only into a reduction in the chances of survival of a given patient but also, to a significant extent, disruption of the functioning of the STEMI treatment system. Unfortunately, the realities of the daily operation of the TTECG system in Poland are different from those recommended in the ESC guidelines. There are no written and legally regulated protocols, as well as a lack of clearly divided regions of geographical responsibility for a given PCI center, which favors the abuse and suboptimal use of the abovementioned telematic systems. Therefore, there is a clear need to further formalize the principles of using telematic systems in modern emergency medicine. Importantly, these rules should be differentiated for each country separately, based on the realities of the health care system. The method of organizing the EMS, the number and availability of CLs, as well as the availability of ICU beds in a given area, should always be considered. Another issue that needs to be clarified is the issue of legal liability. A widespread problem in everyday work is the phenomenon of trying to shift the responsibility for treating a patient to a cardiologist on duty. There are cases of EMS teams refusing to transfer the patient to hospital, explained by the fact that after transmission of the TTECG, the patient was disqualified from transfer to a PCI center. The fact that the patient does not require immediate invasive treatment and transport to the Cl does not mean that the patient does not have a MI; still less does it mean that there is no need to transfer the patient to the hospital. The decision to transfer the patient to the ED should be made by the head of the EMS team each time after carefully collecting the medical history and conducting a physical examination of the patient. The cardiologist on duty only consults the ECG and, in the absence of the features of a STEMI, is not able to determine the necessity or otherwise of transferring the patient to the hospital. Summing up, the application of telematic systems absolutely requires legal regulation as well as further extensive education and making the members of EMS aware of the indications and contraindications for the use of the TTECG system.

## Figures and Tables

**Table 1 medicina-58-00554-t001:** Recommendations for transmission of the TTECG and cooperation between PCI centers and EMS teams.

Recommendations	Class of Recommendation
Clearly defined areas of geographical responsibility of a given CL.	I
A written protocol containing the rules and principles of cooperation when using telematic systems, which all members of the EMS included in the STEMI treatment system are familiar with.	I
Patient with typical chest pain suggestive of acute MI	I
SCA survivors	I
Patient after ICD shock therapy, especially in patients with recurrent arrhythmia, consecutive ICD shocks, reporting stenocardial symptoms, patients with low blood pressure	I
Patient with pulmonary edema of presumed cardiological etiology, accompanied by stenocardia, with symptoms of hemodynamic instability	I
Atypical chest pain in young patients with a potentially minimal risk of MI	IIa
Patient with upper abdominal pain suggesting an abdominal mask of MI	IIa
Patients with hemodynamically unstable supraventricular tachycardia presenting with typical stenocardial symptoms	IIa
Patients with VA (wide QRS complexes) after initial clinical stabilization (restoration of sinus rhythm)	IIa
Patient with recurrent syncope, with typical stenocardial pain and with symptoms of hemodynamic instability	IIa
Patient with bradycardia without typical stenocardial pain (regardless of the mechanism)	IIb
Patient with hemodynamically stable supraventricular arrhythmia (with narrow QRS complexes), who does not present stenocardial pain	IIb
Patient with mild dyspnea, no stenocardial symptoms, no signs of hemodynamic instability	IIb
Patient after syncope, without stenocardia and hemodynamically stable	IIb
Transfer of the patient to the PCI center despite the refusal to admit the patient by the doctor on duty (regardless of the diagnosis of STEMI)	III
Transfer of each patient with chest pain to a PCI center	III
Patients presenting headaches or dizziness, disturbed consciousness and/or general malaise	III
Patient after a chest trauma	III
Patient presenting with abdominal pain, peritoneal signs, abdominal tenderness, and a lack of intestinal peristalsis suggesting an acute condition in the abdominal cavity.	III
Transmission of the TTECG with diagnosed wide-complex tachycardia (suspected ventricular tachycardia)	III
Transfer of a patient with ongoing VA (refractory to treatment) to a distant PCI center	III
Patients in a coma	III
Routine transport of each SCA survivor to a PCI center	III

Abbreviation list: CL—catheterization laboratory, EMS—emergency medical system, STEMI—ST elevation myocardial infarct, MI—myocardial infarct, SCA—sudden cardiac arrest, ICD—implanted cardioverter–defibrillator, VA—ventricular arrhythmias, PCI center—centers performing percutaneous coronary interventions, TTECG—transtelephonic electrocardiography.

## Data Availability

Not applicable.

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
