# Peer review of "Practical Aspects of the Use of Telematic Systems in the Diagnosis of Acute Coronary Syndrome in Poland"

_medicina, 2022, doi:10.3390/medicina58040554_

Round 1

Reviewer 1 Report

Reviewer regarding the manuscript:

Practical aspects of the use of telematic systems in the diagnosis of acute coronary syndromes in Poland.

Gawinski and colleagues report a systematic review of transtelephonic ECG (TTECG) in cardiological emergencies (focusing on ACS). The authors according to their opinion grade the indications for the: 1) effectiveness of TTECG in cardiac emergencies and 2) the mode of collaboration between emergency teams and PCI centers. The indications/conclusions are based on experiences gained in the Polish public health care system.

The paper is interesting but relatively poorly written (needs improvement - especially the English wording).

I would have some comments.

  • The paper is a systematic review. Consequently, it should not follow the typical headings of the regular papers (i.e., Methods and Results etc.). Practically, no real methods or results are presented in this manuscript (because it is an interesting review). I would use heading something like the following: abstract - introduction – aim of the work - cooperation between EMS and PCI center - indications and contraindications for TTECG transmission to a PCI center – conclusions.
  • Table 1. is not needed (it is well-known from the guidelines).
  • Please, provide a legend for Table 2 (give the used abbreviations).
  • Table 2: there is a contradiction regarding SCA survivors. According to the table on one hand it is a class I indication (SCA survivors), but on the other hand in the same table is given as a contraindication (routine transport of each SCA survivors to a PCI center). Please explain.

Author Response

Dear Sir/Madam,

Thank you for taking the time to review our manuscript and all comments! They certainly allowed us to improve our paper. All comments have been deeply analyzed and entered into the manuscript. 

In this place, point by point we would like to answer for your comments and suggestions:

The paper is interesting but relatively poorly written (needs improvement - especially the English wording).

Thank you for that comment. According to your opinion, the article has undergone English language editing (by MDPI). Please find the attached English Editing Certificate.

The paper is a systematic review. Consequently, it should not follow the typical headings of the regular papers (i.e., Methods and Results etc.). Practically, no real methods or results are presented in this manuscript (because it is an interesting review). I would use heading something like the following: abstract - introduction – aim of the work - cooperation between EMS and PCI center - indications and contraindications for TTECG transmission to a PCI center – conclusions.

Thank you for this comment. According to your recommendations, we changed the structure of the paper and used the headings you suggested.

 Table 1. is not needed (it is well-known from the guidelines).

Thank you for this comment. According to your recommendation, we have removed this table from the manuscript.

Please, provide a legend for Table 2 (give the used abbreviations).

Thank you very much for this suggestion. We added a legend to table 1 with the list of abbreviations. 

Table 2: there is a contradiction regarding SCA survivors. According to the table on one hand it is a class I indication (SCA survivors), but on the other hand in the same table is given as a contraindication (routine transport of each SCA survivors to a PCI center). Please explain.

Thank you for this valuable insight. Not every SCA survivors has to be routinely transported directly from the scene to a PCI center (suggested class of recommendations: III), but we should transmit the TTECG of every SCA survivors to a PCI center (suggested class of recommendations: I).
The revised manuscripts contain an additional sentence explaining this issue in details.

Once again we would like to sincerely thank you for a very comprehensive, insightful and in many places accurate review. All of the above comments are highly valuable for the comprehensiveness of the paper and our own scientific development.

Sincerely yours,

Authors

Reviewer 2 Report

Lukasz Gawinski et al wrote guidelines when to activate an ECG teletransmission system by EMS in Poland. While the recommendations seem solid and backed up by literature, the authors have to make clear under which settings they are valid. Therefore, I invite the authors to add a section briefly describing the EMS system in Poland. Furthermore, they should clearly state that those recommendations are only valid in the case of ECG teletransmission to identify STEMI or high-risk NSTEMI patients. I do not understand the recommendation that an NSTEMI patient should not be transferred to the PCI centre in case of a PCI and a non-PCI centre being at similar distance. In my opinion, it may ease the further management of the patient. Finally, in some situations ECG teletransmission may be of value even in certain situations with a low likelihood of myocardial infarction and proposed class III recommendation. Imagine the following situation: A patient could have chest pain due to STEMI, may lose consciousness while driving and gets therefore involved in a car accident with chest trauma. The recommendations proposed by the authors would prohibit ECG teletransmission, even if the patient would benefit from acute PCI.

Author Response

Dear Sir/Madam,

Thank you for taking the time to review our manuscript and all comments! They certainly allowed us to improve our paper. All comments have been deeply analyzed and entered into the manuscript. 

In this place, point by point we would like to answer for your comments and suggestions:

While the recommendations seem solid and backed up by literature, the authors have to make clear under which settings they are valid. Therefore, I invite the authors to add a section briefly describing the EMS system in Poland. Furthermore, they should clearly state that those recommendations are only valid in the case of ECG teletransmission to identify STEMI or high-risk NSTEMI patients.

Thank you for that comment. These are very valuable insights. Due to the fact that in my previous work I described in great detail the structure and functioning of EMS in Poland, I allowed myself to quote the article in the revised manuscript: 13. Gawinski, L.P .; Kozlowski, R. Telematic Support of Management Processes in Diagnosis and Treatment of Acute Myocardial Infarction in Poland. In Research and the Future of Telematics; Mikulski, J., Ed .; Communications in Computer and Information Science; Springer International Publishing: Cham, 2020; Vol. 1289, pp. 443–455 ISBN 978-3-030-59269. Additionally, a sentence has been included indicating that the main application for the ECG teletransmission system is the diagnostic process of patients with STEMI and high-risk NSTEMI.

I do not understand the recommendation that an NSTEMI patient should not be transferred to the PCI centre in case of a PCI and a non-PCI centre being at similar distance. In my opinion, it may ease the further management of the patient.

Thank you for this comment, however I haven’t found such recommendation in my manuscript. When analyzing the applications of the ECG teletransmission system, we stated unequivocally that patients with STEMI and patients with NSTEMI (meeting certain high risk of NSTEMI criteria) should be transported directly to the PCI center. The remaining patients can be transported to the nearest hospital, regardless of whether it has a PCI center in its structure. Of course, if the nearest hospital is a hospital with a PCI center, the patient should be transported there (although it seems that it does not require the implementation of an immediate invasive strategy).

“The remaining patients may be transported to the ED of the nearest hospital, where they will be given the necessary medical assistance and appropriate diagnostic procedures will be conducted. After confirming NSTEMI, in accordance with the risk assessment carried out, these patients can be transported to a PCI center within appropriate timeframes in accordance with the ESC guidelines. Of course, this rule does not apply to patients for whom the hospital with the PCI center is the nearest hospital”

Finally, in some situations ECG teletransmission may be of value even in certain situations with a low likelihood of myocardial infarction and proposed class III recommendation. Imagine the following situation: A patient could have chest pain due to STEMI, may lose consciousness while driving and gets therefore involved in a car accident with chest trauma. The recommendations proposed by the authors would prohibit ECG teletransmission, even if the patient would benefit from acute PCI.

Thank you very much for this suggestion. Providing class III recommendations for: ECG teletransmission in a patient after a chest trauma, I was guided by the fact that EMS members very often associate the fact of chest trauma with the possibility of a secondary myocardial infarct and uncritically and routinely transmit ECG to the PCI center overload with the work of doctors on duty. One of the goals of this paper was to identify the typical and most common indications for an ECG teletransmission, thus limiting the number of unnecessary ECG teletransmissions. Chest trauma followed by myocardial infarction is very rare. Much more often, a chest trauma causes a contusion of the heart (which is not an typical indication for coronary intervention). In my professional career, I have experienced only one case of a chest injury resulting in a myocardial infarction (a knife attack in the chest followed physical rupture of the right coronary artery). The patient underwent a cardiac surgery of suturing the walls of the heart followed by CABG. Therefore, the recommendation I have mentioned specifically pointed at chest trauma. In the situation presented by the reviewer, we are dealing primarily with myocardial infarction complicated probably by syncope and subsequent chest injury (another part of the body could have been damaged then). In the presented case, the guidelines for syncope should be followed, in which I assessed the validity of ECG teletransmission as IIa or IIb class. Whenever there is a reasonable suspicion that the trauma is secondary to loss of consciousness, and hence to myocardial infarction, ECG transmission to a PCI center should / may be considered.

Once again we would like to sincerely thank you for a very comprehensive, insightful and in many places accurate review. All of the above comments are highly valuable for the comprehensiveness of the paper and our own scientific development.

Sincerely yours,

Authors